# Unravelling the novel genetic diversity and marker-trait associations of corn leaf aphid resistance in wheat using microsatellite markers

Jayant Yadav[1,2], Poonam Jasrotia[1]*, Maha Singh Jaglan[2], Sindhu Sareen[1], Prem Lal Kashyap[1], Sudheer Kumar[1], Surender Singh Yadav[2], Gyanendra Singh[1], Gyanendra Pratap Singh[1,3]

**1** ICAR- Indian Institute of Wheat and Barley Research, Karnal, Haryana, India, **2** CCS Haryana Agricultural University, Hisar, Haryana, India, **3** ICAR- National Bureau of Plant Genetic Resources, New Delhi, India

* poonam.jasrotia@icar.gov.in, poonamjasrotia@gmail.com

## Abstract

The study was conducted to identify novel simple sequence repeat (SSR) markers associated with resistance to corn aphid (CLA), *Rhopalosiphum maidis* L. in 48 selected bread wheat (*Triticum aestivum* L.) and wild wheat (*Aegilops* spp. & *T. dicoccoides*) genotypes during two consecutive cropping seasons (2018–19 and 2019–20). A total of 51 polymorphic markers containing 143 alleles were used for the analysis. The frequency of the major allele ranged from 0.552 (*Xgwm113*) to 0.938 (*Xcfd45*, *Xgwm194* and *Xgwm526*), with a mean of 0.731. Gene diversity ranged from 0.116 (*Xgwm526*) to 0.489 (*Xgwm113*), with a mean of 0.354. The polymorphic information content (PIC) value for the SSR markers ranged from 0.107 (*Xgwm526*) to 0.370 (*Xgwm113*) with a mean of 0.282. The results of the STRUCTURE analysis revealed the presence of four main subgroups in the populations. Analysis of molecular variance (AMOVA) showed that the between-group difference was around 37 per cent of the total variation contributed to the diversity by the whole germplasm, while 63 per cent of the variation was attributed between individuals within the group. A general linear model (GLM) was used to identify marker-trait associations, which detected a total of 23 and 27 significant new marker-trait associations (MTAs) at the p < 0.01 significance level during the 2018–19 and 2019–20 crop seasons, respectively. The findings of this study have important implications for the identification of molecular markers associated with CLA resistance. These markers can increase the accuracy and efficiency of aphid-resistant germplasm selection, ultimately facilitating the transfer of resistance traits to desirable wheat genotypes.

## Introduction

Aphids are considered as serious insect pests of wheat crops worldwide [1, 2]. More than 11 aphid species have been reported to cause damage to wheat crop. However, in India, *Sitobion avenae* (Fab.), *S. miscanthi* (Takahashi), *Rhopalosiphum padi* (L.) and *R. maidis* (Fitch) are the

**Data Availability Statement:** All relevant data are within the paper.

**Funding:** Yes. The funding for conducting the experiment was provided under the Institute project "Management of wheat insect-pests through climate-smart pest management strategies" of ICAR-ICAR- Indian Institute of Wheat and Barley Research, Karnal 132001, Haryana, India.

**Competing interests:** The authors have declared that no competing interests exist.

four predominant aphid species and together these are called as wheat aphid complex [3–5]. Amongst those, the corn leaf aphid (CLA), *Rhopalosiphum maidis* is considered as economically significant aphid species of North-Western plains of India [6, 7]. CLA feed on wheat plants by sucking sap from the leaves, stems, and grains. The peculiar characteristic of the aphids that make them destructive is their high multiplication rate. Their damage by aphids leads to includes stunted growth, reduced tillering, yellowing of leaves, curling, and deformation of wheat heads. Furthermore, aphids excrete honeydew that acts as a substrate for the growth of sooty mould, further affecting plant health [8, 9]. The aphid damage on wheat initiates from seedling stage onwards at during October-November, however it is difficult to detect because of their small size and green coloration. The peak population of aphid was usually reported during February to March [10, 11]. Under severe incidence of aphids, wheat can suffer extensive yield losses, ranging from 20 per cent to 30 per cent [12, 13]. Early infestations at ear head emergence stage can result in yield reductions of up to 14 per cent, with the degree of yield loss decreasing as infestations occur later in the crop growth cycle [11, 14].

Various management strategies such as proper crop rotation, timely sowing, resistant varieties and chemicals are used to control aphid infestations in wheat. Control of aphids using systemic insecticides is expensive and poses risks to human health as well as to environment [15, 16]. On the other hand, the strategy of breeding for resistant cultivars offers a valuable and environmentally friendly approach for controlling aphid damage, reducing pesticide use, preserving natural ecosystems, and enhancing crop productivity in a sustainable manner. Breeding programs in India have focused mainly on developing aphid-resistant wheat varieties through conventional breeding techniques [17]. A recent approach, Marker-assisted selection (MAS) uses molecular markers for identification and incorporation of aphid resistance genes into high-yielding wheat varieties. This approach allows breeders to select for aphid resistance more efficiently, accurately and reduce the reliance on chemical control methods [18–20]. Earlier, a series of resistant wheat varieties were introduced, but due to the formation of new biotypes of the pest, focus on resistance research was shifted [7, 21–24].

In the context of developing resistant varieties through MAS, QTL's linked to insect resistance genes can be identified using Simple-Sequence Repeats (SSRs), By analyzing the genetic profiles of different plants, breeders can identify individuals that carry the desired insect resistance genes based on their SSR marker patterns. This information allows breeders to select and prioritize plants with the highest likelihood of possessing the desired resistance traits, improving the efficiency of the breeding process [25]. SSRs have gained popularity in genetic research due to their characteristics. They are highly polymorphic, meaning they exhibit a high level of variation within a population. This makes them valuable in distinguishing different genotypes and identifying specific alleles associated with target traits. SSRs are also co-dominant markers, which means that both alleles at a given locus can be distinguished, allowing for more precise genetic analysis [26]. SSR markers have been previously employed in many plant species such as rice [27], maize [28], soybean [29] and wheat [30–32]. These markers have been found to be more variable than other marker systems like Restriction Fragment Length Polymorphisms (RFLPs). This high variability enhances their effectiveness in uncovering genetic variation within a population and identifying specific genes or alleles of interest. Besides, SSR markers have shown efficiency in species those having relatively low levels of intraspecific polymorphism. Hexaploid wheat, for example, is a self-pollinating species with limited genetic diversity. SSR markers have been successfully employed in breeding platforms of wheat, aiding in the identification and selection of desirable trait [33, 34].

The use of molecular genetic diversity and marker-trait associations has proven to be valuable in choosing better parental materials for breeding aphid-resistant wheat varieties [7]. In this context, SSR markers have played a crucial role in various aspects of wheat breeding, such

as genomic mapping, marker-assisted breeding, genetic analyses, and assessing the diversity and polymorphism of wheat germplasm [35]. A large number of SSR markers have been extensively utilized in wheat improvement programs. They have been utilized in marker-assisted selection and genomic mapping analyses, and evaluating the diversity and polymorphism of wheat germplasm [36–42]. Association mapping, a technique that examines the association between molecular markers and phenotypic traits in elite germplasm, has proven to be a valuable tool complementing QTL (Quantitative Trait Locus) studies and marker-assisted selection efforts [43, 44]. These advancements have significantly contributed to interdisciplinary efforts in plant resistance, leading to improved pest management and sustainable food production. Identifying both morphological and molecular markers associated with aphid resistance enables more accurate and efficient selection of resistant germplasm, facilitating the transfer of resistance traits to desirable wheat genotypes. It has been reported that landraces and wild relatives of wheat have significant level of resistance against aphids [45]. Moreover, by identifying tightly linked markers to previously reported QTLs governing aphid resistance, breeding programs can enhance their effectiveness in developing aphid-resistant wheat plants [46, 47].

Considering these factors, the present study aims to investigate the genetic diversity and marker-trait associations for aphid resistance in wheat using microsatellite markers. This research will provide valuable insights for crop improvement programs focused on breeding aphid-resistant wheat varieties, ultimately contributing to sustainable and resilient wheat production.

## Materials and methods

### Germplasm details and phenotyping set-up for determining aphid resistant response

The experimental material for the proposed study consisted of a total of 48 aphid tolerant wheat genotypes, which included 25 bread wheat (*T. aestivum*) varieties and 23 wild wheat (*Aegilops spp.* & *T. dicoccoides*) genotypes (Table 1). These genotypes were screened in a net house for recording the aphid resistance response.

For screening, each genotype was sown in one-meter rows and a row spacing of 25 cm was kept. The experiment followed a randomized block design (RBD) and was conducted during the 2018–19 and 2019–20 *Rabi* seasons (November-April). There were three replications and each replication has two rows per replication. A recommended set of agronomic practices was followed to ensure the healthy growth of the wheat crop except the spray of any pesticide was avoided. To initiate aphid infestation, small pots containing 100 aphids were placed between all the rows to build aphid pressure for screening. The aphids used included alates (winged aphids), apterae adults (wingless adults), and nymphs.

The extent of the aphid infestation was determined by recording the number of aphids on five randomly selected shoots in each row. All aphid stages i.e. alates, apterae adults, and nymphs were counted. The mean number of aphid population per shoot was calculated for each replication. Subsequently, the mean of the three replications was used as the phenotypic data, representing the mean number of aphids per shoot for each wheat genotype. This phenotyping approach allowed for the evaluation of aphid resistance in the wheat genotypes and provided data on the mean number of aphids per plant, which was utilized for further analysis and identification of resistant germplasm.

### Molecular characterization of *R. maidis* resistance

Molecular characterization of *R. maidis* resistance in wheat varieties and wild wheat genotypes was carried out during 2020–21 *Rabi* season. For genotyping, leaf samples from one-month-

**Table 1. List of the bread wheat and wild wheat genotypes used in the study.**

| Sr. No. | Genotype code | Bread wheat (*Triticum aestivum*) genotypes | Sr. No. | Genotype code | Wild wheat species | Accession numbers |
|---------|---------------|---------------------------------------------|---------|---------------|--------------------|--------------------|
| 1. | G5 | DBW 17 | 1. | W4 | *Aegilops speltoides* | 3595 |
| 2. | G11 | DBW 71 | 2. | W5 | *Aegilops speltoides* | 3581 |
| 3. | G17 | DBW 88 | 3. | W9 | *Aegilops speltoides* | 3584 |
| 4. | G9 | DBW 90 | 4. | W12 | *Aegilops tauschii* | 3758 |
| 5. | G1 | DBW 187 | 5. | W14 | *Aegilops tauschii* | 14336 |
| 6. | G10 | WH 283 | 6. | W25 | *Aegilops peregrina* | PI 604173 |
| 7. | G12 | WH 542 | 7. | W26 | *Aegilops peregrina* | PI 604172 |
| 8. | G14 | WH 1080 | 8. | W37 | *Aegilops tauschii* | 15 |
| 9. | G25 | WH 1105 | 9. | W47 | *Aegilops tauschii* | 13762 |
| 10. | G18 | WH 1124 | 10. | W51 | *Aegilops tauschii* | 9787 |
| 11. | G21 | WH 1142 | 11. | W59 | *Aegilops peregrina* | 54 |
| 12. | G22 | PBW 343 | 12. | W68 | *Aegilops tauschii* | 9788 |
| 13. | G16 | PBW 373 | 13. | W79 | *Aegilops tauschii* | 9798 |
| 14. | G19 | PBW 396 | 14. | W87 | *Aegilops peregrina* | PI 604186 |
| 15. | G15 | PBW 550 | 15. | W91 | *Triticum dicoccoides* | 13993 |
| 16. | G20 | PBW 644 | 16. | W98 | *Aegilops speltoides* | 3599 |
| 17. | G8 | HD 2967 | 17. | W103 | *Aegilops speltoides* | 3761 |
| 18. | G6 | HD 3043 | 18. | W111 | *Aegilops peregrina* | PI 604162 |
| 19. | G24 | RAJ 3077 | 19. | W115 | *Aegilops peregrina* | PI 604185 |
| 20. | G13 | UP 2425 | 20. | W121 | *Aegilops speltoides* | 3590 |
| 21. | G7 | WHD 943 | 21. | W123 | *Aegilops speltoides* | 3596 |
| 22. | G23 | HD 3086 | 22. | W145 | *Aegilops tauschii* | 13764 |
| 23. | G4 | A-9-30-1 | 23. | W164 | *Aegilops tauschii* | 13765 |
| 24. | G2 | DBW 222 | | | | |
| 25. | G3 | DBW 303 | | | | |

old seedlings of wheat and wild wheat genotypes in duplicate were collected. These samples were placed in liquid nitrogen and kept at -20˚C until DNA extraction was carried out. Before performing DNA extraction, these samples were put in liquid nitrogen and stored at a temperature of -20˚C. A modified CTAB (Cetyltrimethylammonium bromide) extraction procedure [48] was used to get genomic DNA using from the 30-day-old leaves. The $A_{260}/A_{280}$ absorbance ratio was measured using a nanodrop/spectrophotometer to evaluate the amount and quality of the extracted DNA. Additionally, agarose gel electrophoresis was used to more thoroughly assess the DNA's purity. The bread wheat and wild wheat genotypes were genotyped using a set of 51 SSR markers (Table 2) for molecular screening and downstream analysis. The PCR reaction mixture was optimized and consisted of 10 μl total volume, including 50 ng of DNA, 0.5 μl of primers, 3 μl of nuclease-free water and 5 μl of MasterMix (GoTaq® Green Master Mix). Using a 2 per cent agarose gel in 1.0X TBE buffer at 4 V/cm, PCR products were then separated by gel electrophoresis. After the electrophoretic gels were stained with ethidium bromide at a concentration of 0.5 mg/ml, the ensuing DNA banding patterns of the 48 genotypes were observed under UV light.

## SSR marker evaluation and genomic diversity assessment

Each position with an amplified band received a score of 1 for presence and 0 for absence. Based on the migration of the amplified bands in relation to the industry-standard 100-bp DNA ladder (Promega), the size of the bands was further estimated (in nucleotide base pairs).

**Table 2. Brief description of SSR primers used during the study.**

| Sr. No. | Marker Name | Chromosomal location | Forward and Reverse sequences | Annealing temperature (˚C) | No. of alleles |
|---|---|---|---|---|---|
| 1. | *Xpsp3000* | *1B, 1D* | F-5'-GCAGACCTGTGTCATTGGTC-3' | 60 | 3 |
| | | | R-5'-GATATAGTGGCAGCAGGATACG-3' | | |
| 2. | *Xgwm44* | *7D* | F-5' GTTGAGCTTTTCAGTTCGGC 3' | 60 | 3 |
| | | | R-5' ACTGGCATCCACTGAGCTG 3' | | |
| 3. | *Xpsp3079* | *6B, 4D, 7D* | F-5'-CGAAAGGCTAGAAAACAGGAACG-3' | 60 | 2 |
| | | | R-5'-CTCGCGATGTTGCCCCAGCG-3' | | |
| 4. | *Xgwm111* | *7B, 7D* | F-5' TCTGTAGGCTCTCTCCGACTG 3' | 55 | 4 |
| | | | R-5' ACCTGATCAGATCCCACTCG 3' | | |
| 5. | *Xgwm106* | *1A, 1B, 1D* | F-5' CTGTTCTTGCGTGGCATTAA 3' | 60 | 2 |
| | | | R-5' AATAAGGACACAATTGGGATGG 3' | | |
| 6. | *Xgwm337* | *1B, 1D* | F-5' CCTCTTCCTCCCTCACTTAGC 3' | 60 | 3 |
| | | | R-5' TGCTAACTGGCCTTTGCC 3' | | |
| 7. | *Xgwm642* | *1D* | F-5' ACGGCGAGAAGGTGCTC 3' | 55 | 2 |
| | | | R-5' CATGAAAGGCAAGTTCGTCA 3' | | |
| 8. | *Xgwm136* | *1A* | F-5' GACAGCACCTTGCCCTTTG 3' | 55 | 3 |
| | | | R-5' CATCGGCAACATGCTCATC 3' | | |
| 9. | *Xcfd45* | *6D* | F-5' TCTCTCCAGTTGCTCCTCGT 3' | 55 | 2 |
| | | | R-5' ATGTGGAACCGGTCTACTCG 3' | | |
| 10. | *Xgwm473* | *2A* | F-5' TCATACGGGTATGGTTGGAC 3' | 54 | 2 |
| | | | R-5' CACCCCCTTGTTGGTCAC 3' | | |
| 11. | *Xgwm635* | *7A* | F-5' TTCCTCACTGTAAGGGCGTT 3' | 60 | 2 |
| | | | R-5' CAGCCTTAGCCTTGGCG 3' | | |
| 12. | *Xgwm174* | *5D* | F-5' GGGTTCCTATCTGGTAAATCCC 3' | 55 | 4 |
| | | | R-5' GACACACATGTTCCTGCCAC 3' | | |
| 13. | *Xpsp3029* | *2A* | F-5' CCATCGATGAGGATCTCCTCGGGCA 3' | 60 | 2 |
| | | | R-5' GCAACAGGACCATGGTCG 3' | | |
| 14. | *Xgwm260* | *7A* | F-5' GCCCCCTTGCACAAATC 3' | 55 | 2 |
| | | | R-5' CGCAGCTACAGGAGGCC 3' | | |
| 15. | *Xgwm121* | *5D, 7D* | F-5' TCCTCTACAAACAAACACAC 3' | 50 | 4 |
| | | | R-5' CTCGCAACTAGAGGTGTATG 3' | | |
| 16. | *Xgwm148* | *2B* | F-5' GTGAGGCAGCAAGAGAGAAA 3' | 60 | 3 |
| | | | R-5' CAAAGCTTGACTCAGACCAAA 3' | | |
| 17. | *Xgwm495* | *4B* | F-5' GAGAGCCTCGCGAAATATAGG 3' | 60 | 2 |
| | | | R-5' TGCTTCTGGTGTTCCTTCG 3' | | |
| 18. | *Xgwm265* | *2A, 4A* | F-5' TGTTGCGGATGGTCACTATT 3' | 55 | 4 |
| | | | R-5' GAGTACACATTTGGCCTCTGC 3' | | |
| 19. | *Xgwm2* | *3A* | F-5' CTGCAAGCCTGTGATCAACT 3' | 50 | 5 |
| | | | R-5' CATTCTCAAATGATCGAACA 3' | | |
| 20. | *Xgwm391* | *3A* | F-5' ATAGCGAAGTCTCCCTACTCCA 3' | 55 | 4 |
| | | | R-5' ATGTGCATGTCGGACGC 3' | | |
| 21. | *Xgwm165* | *4A, 4B, 4D* | F-5' TGCAGTGGTCAGATGTTTCC 3' | 60 | 3 |
| | | | R-5' CTTTTCTTTCAGATTGCGCC 3' | | |
| 22. | *Xgwm637* | *4A* | F-5' AAAGAGGTCTGCCGCTAACA 3' | 60 | 3 |
| | | | R-5' TATACGGTTTTGTGAGGGGG 3' | | |
| 23. | *Xgwm304* | *5A* | F-5' AGGAAACAGAAATATCGCGG 3' | 55 | 2 |
| | | | R-5' AGGACTGTGGGGAATGAATG 3' | | |

*(Continued)*

**Table 2.** (*Continued*)

| Sr. No. | Marker Name | Chromosomal location | Forward and Reverse sequences | Annealing temperature (˚C) | No. of alleles |
|---|---|---|---|---|---|
| 24. | *Xgwm276* | *7B* | F-5' ATTTGCCTGAAGAAAATATT 3' | 55 | 3 |
| | | | R-5' AATTTCACTGCATACACAAG 3' | | |
| 25. | *Xgwm153* | *1B* | F-5' GATCTCGTCACCCGGAATTC 3' | 60 | 3 |
| | | | R-5' TGGTAGAGAAGGACGGAGAG 3' | | |
| 26. | *Xgwm526* | *2A, 2B, 7A, 7B* | F-5' CAATAGTTCTGTGAGAGCTGCG 3' | 55 | 3 |
| | | | R-5' CCAACCCAAATACACATTCTCA 3' | | |
| 27. | *Xgwm77* | *3B* | F-5' ACAAAGGTAAGCAGCACCTG 3' | 60 | 3 |
| | | | R-5' ACCCTCTTGCCCGTGTTG 3' | | |
| 28. | *Xgwm299* | *3B, 3R* | F-5' ACTACTTAGGCCTCCCGCC 3' | 55 | 3 |
| | | | R-5' TGACCCACTTGCAATTCATC 3' | | |
| 29. | *Xgwm113* | *4B* | F-5' ATTCGAGGTTAGGAGGAAGAGG 3' | 55 | 2 |
| | | | R-5' GAGGGTCGGCCTATAAGACC 3' | | |
| 30. | *Xgwm537* | *7B* | F-5' ACATAATGCTTCCTGTGCACC 3' | 60 | 2 |
| | | | R-5' GCCACTTTTGTGTCGTTCCT 3' | | |
| 31. | *Xgwm540* | *5B* | F-5' TCTCGCTGTGAAATCCTATTTC 3' | 55 | 2 |
| | | | R-5' AGGCATGGATAGAGGGGC 3' | | |
| 32. | *Xgwm335* | *5B* | F-5' CGTACTCCACTCCACACGG 3' | 55 | 5 |
| | | | R-5' CGGTCCAAGTGCTACCTTTC 3' | | |
| 33. | *Xgwm508* | *6B* | F-5' GTTATAGTAGCATATAATGGCC 3' | 50 | 3 |
| | | | R-5' GTGCTGCCATGATATTT 3' | | |
| 34. | *Xgwm193* | *6B* | F-5' CTTTGTGCACCTCTCTCTCC 3' | 60 | 2 |
| | | | R-5' AATTGTGTTGATGATTTGGGG 3' | | |
| 35. | *Xgwm146* | *7B* | F-5' CCAAAAAAACTGCCTGCATG 3' | 60 | 3 |
| | | | R-5' CTCTGGCATTGCTCCTTGG 3' | | |
| 36. | *Xgwm210* | *2B* | F-5' TGCATCAAGAATAGTGTGGAAG 3' | 60 | 2 |
| | | | R-5' TGAGAGGAAGGCTCACACCT 3' | | |
| 37. | *Xgwm301* | *2D* | F-5' GAGGAGTAAGACACATGCCC 3' | 55 | 5 |
| | | | R-5' GTGGCTGGAGATTCAGGTTC 3' | | |
| 38. | *Xgwm314* | *3D* | F-5' AGGAGCTCCTCTGTGCCAC 3' | 55 | 2 |
| | | | R-5' TTCGGGACTCTCTTCCCTG 3' | | |
| 39. | *Xgwm383* | *3B, 3D* | F-5' ACGCCAGTTGATCCGTAAAC 3' | 60 | 2 |
| | | | R-5' GACATCAATAACCGTGGATGG 3' | | |
| 40. | *Xgwm194* | *4D* | F-5' GATCTGCTCTACTCTCCTCC 3' | 50 | 2 |
| | | | R-5' CGACGCAGAACTTAAACAAG 3' | | |
| 41. | *Xgwm192* | *5D* | F-5' GGTTTTCTTTCAGATTGCGC 3' | 60 | 3 |
| | | | R-5' CGTTGTCTAATCTTGCCTTGC 3' | | |
| 42. | *Xcfd14* | *7D* | F-5' CCACCGGCCAGAGTAGTATT 3' | 60 | 2 |
| | | | R-5' TCCTGGTCTAACAACGAGAAGA 3' | | |
| 43. | *Xcfd68* | *7D* | F-5' TTTGCAGCATCACACGTTTT 3' | 60 | 2 |
| | | | R-5' AAAATTGTATCCCCCGTGGT 3' | | |
| 44. | *Xpsp3200* | *6D* | F-5' GTTCTGAAGACATTACGGATG 3' | 61 | 2 |
| | | | R-5' GAGAATAGCTGGTTTTGTGG 3' | | |
| 45. | *Xbarc128* | *2B* | F-5' GCGGGTAGCATTTATGTTGA 3' | 52 | 2 |
| | | | R-5' CAAACCAGGCAAGAGTCTGA 3' | | |
| 46. | *Xbarc148* | *1A* | F-5' GCGCAACCACAATGTATGCT 3' | 52 | 3 |
| | | | R-5' GGGGTGTTTTCCTATTTCTT 3' | | |

(*Continued*)

**Table 2.** (Continued)

| Sr. No. | Marker Name | Chromosomal location | Forward and Reverse sequences | Annealing temperature (˚C) | No. of alleles |
|---|---|---|---|---|---|
| 47. | Xbarc171 | 6A | F-5' GCGGGGTCATCTTAGTAACTCAAATA 3' | 50 | 3 |
| | | | R-5' ACTGTCAACGTTGGTTCACATTCA 3' | | |
| 48. | Xbarc172 | 7D | F-5' GCGAAATGTGATGGGGTTTATCTA 3' | 50 | 2 |
| | | | R-5' GCGATTTGATTTAACTTTAGCAGTGAG 3' | | |
| 49. | Xbarc17 | 1A | F-5' GCGCAACATATTCAGCTCAACA 3' | 50 | 4 |
| | | | R-5' TCCACATCTCGTCCCTCATAGTTTG 3' | | |
| 50. | Xbarc126 | 7D | F-5' CCATTGAAACCGGATTTGAGTCG 3' | 52 | 3 |
| | | | R-5' CGTTCCATCCGAAATCAGCAC 3' | | |
| 51. | Xbarc214 | 7D | F-5' CGCTTTCGGGACAGTGAAGGTGTAT 3' | 52 | 4 |
| | | | R-5' CGGTACGCGCGAGGAGGAAGAAGG 3' | | |
| | | | | **Total number of alleles** | **143** |
| | | | | **Average number of alleles per marker** | **2.804** |

In order to do a preliminary statistical analysis of the genotypes by POWERMARKER V 3.25, the molecular weights of SSR products (measured in base pairs) were determined [49]. For each marker, calculations were made to determine the total number of alleles, major allele frequency, polymorphism information content (PIC) values, gene diversity, etc.

## Population structure analysis

Using the programme STRUCTURE V 2.3.4 [50], the population structure of the 48 wheat genotypes was examined. For the genotypes under study, an admixture model was used to calculate the K value, which represents the number of subpopulations. For the analysis, 10 iterations were done for each value of K between 1 and 10.

A burn-in length of 10,000 and a specified number of replications were employed as additional parameters. These runs enabled the genotype-level assessment of population sub-structure and individual ancestry within the genotypes. The optimal number of sub-populations (K value) was measured using the procedure described by [51]. This approach makes use of the K statistic, which gauges the rate of change in the data's log probability between successive K values. The uppermost hierarchical level of the structure is most precisely represented by the graph's highest value, which is obtained by graphing K values. K value was calculated by dividing the standard deviation of L(K) by the mean of the absolute differences between successive probability values of K.

To assist in this analysis, STRUCTURE HARVESTER software tool [52] was utilized. This programme creates K plots for the genotypes under study, enabling the selection of the ideal K value. Additionally, phylogenetic tree construction was carried out using DARwin 6 software to better analyse the population structure. Using the Past 4.08 programme, Principal Coordinate Analysis (PCoA) was carried out.

## Analysis of Molecular Variance (AMOVA)

Using GenAlex version 6.5 [53], the molecular variation of the bread wheat and wild wheat genotypes was evaluated. Following the procedure outlined by [54], the parameters computed for genetic diversity comprised the total number of alleles per locus (Na), number of effective alleles per locus (Ne), Shannon's information index (I), observable gene diversity (h), unbiased gene diversity (uh).

### Linkage Disequilibrium (LD) and association mapping

The TASSEL 5 programme was used to evaluate Linkage Disequilibrium (LD) [55]. When two or more loci on the same chromosome or on distinct chromosomes co-segregate or associate non-randomly, this is referred to as LD. To quantify LD, the software calculated the allele frequency correlation ($r^2$) between pairs of markers that were located on the same chromosome. This allowed for the assessment of the degree of linkage between alleles and the identification of regions with high LD. Using a Generalised Mixed Model (GLM) in TASSEL 5, association mapping was carried out for determining relationships between genetic markers and phenotypes of interest. The significance threshold for declaring marker-trait associations (MTAs) was set at -log10(p-value) $\leq$ 2. The depiction of the significant MTAs across the genome was done by plotting the negative logarithm of the p-values on the y-axis against the marker locations on the x-axis. Additionally, utilising marker genotype data based on the VanRaden approach, a marker-based kinship matrix (K) was created. This kinship matrix helps account for relatedness among individuals in the association mapping analysis, reducing the potential for spurious associations. For statistical analyses and data visualization, SPSS version 23.0 [56] and OriginPro 2018 [57] softwares were used, respectively.

## Results

### Phenotyping for aphid resistance

Phenotyping data for determining aphid resistance showed varied response on selected bread wheat and wild wheat genotypes. During the 2018–19 growing season, it was observed that the genotype A-9-30-1 had the highest average number of aphids per tiller (20.59), whereas the genotype DBW 303 had the lowest average number (0.77 aphids/tiller). The same trend was recorded during the 2019–20 with genotype A-9-30-1 having the highest average number of aphids per tiller (21.0 aphids/tiller), whereas genotype DBW 303 had the lowest average (1.04 aphids/tiller). Amongst wild wheat genotypes, the highest mean number of aphids per tiller was recorded on the genotype 13993 (27.80) whereas minimum was observed in the genotype 3590 (11.61) during 2018–19. Similar trend was observed during 2019–20 with genotype 13993 (34.20) having the highest mean number of aphids per tiller recorded whereas lowest was observed on the genotype 3590 (13.07) (Table 3).

### Molecular characterization of *R. maidis* resistance

For the molecular characterization of *R. maidis* resistance, 51 polymorphic markers in all were utilized on 48 genotypes including 25 bread and 23 wild wheat genotypes. With an average of 7.29 markers per chromosome, these markers were found to be distributed across 7 chromosomes. A total of 143 unique alleles among the 51 polymorphic markers were identified. The average number of alleles per marker was 2.804, with allele counts ranging from 2 to 5. This range of alleles indicates the genetic diversity captured by the markers and allows for further analysis of allelic variation and its relationship with *R. maidis* resistance (Table 2).

### Genetic diversity

In the current study, the major allele frequency (MAF) values were analyzed for the 51 polymorphic markers. The MAF values, which represent the frequency of the most prevalent allele at each marker locus within the studied wheat genotypes, varied from 0.552 to 0.938 with a maximum average value of 0.733. The marker *Xcfd45* had the highest MAF of 0.938, indicating that the most common allele at this marker was present in a significant proportion of the genotypes.

**Table 3. Phenotyping of selected bread wheat and wild wheat genotypes under net house for determining aphid resistance response.**

| Sr. No. | Bread wheat genotypes | Mean no. of aphids/tiller | | | Wild wheat genotypes | Mean no. of aphids/tiller | | |
|---|---|---|---|---|---|---|---|---|
| | | 2018–19 | 2019–20 | Overall Mean | | 2018–19 | 2019–20 | Overall Mean |
| 1. | DBW-187 | 1.34 | 1.50 | 1.42 | 9788 | 24.19 | 26.52 | 25.36 |
| 2. | DBW-222 | 1.71 | 1.91 | 1.81 | 3590 | 11.61 | 14.52 | 13.07 |
| 3. | DBW-303 | 0.77 | 1.04 | 0.91 | 13764 | 11.92 | 16.00 | 13.96 |
| 4. | A-9-30-1 | 20.59 | 21.00 | 20.79 | 3596 | 13.08 | 18.42 | 15.75 |
| 5. | DBW 17 | 10.43 | 11.68 | 11.06 | 3761 | 19.10 | 24.40 | 21.75 |
| 6. | HD 3043 | 6.28 | 6.68 | 6.48 | 3599 | 15.53 | 20.68 | 18.11 |
| 7. | WHD 943 | 17.56 | 17.72 | 17.64 | 13993 | 27.80 | 34.20 | 31.00 |
| 8. | HD 2967 | 17.39 | 17.77 | 17.58 | 9787 | 16.48 | 20.96 | 18.72 |
| 9. | DBW 90 | 5.52 | 5.80 | 5.66 | 3595 | 16.60 | 23.81 | 20.21 |
| 10. | WH 283 | 0.84 | 1.08 | 0.96 | 3581 | 16.79 | 19.26 | 18.02 |
| 11. | DBW 71 | 11.82 | 13.31 | 12.57 | 3584 | 15.40 | 18.81 | 17.11 |
| 12. | WH 542 | 15.86 | 16.82 | 16.34 | 3758 | 15.59 | 21.57 | 18.58 |
| 13. | UP 2425 | 2.96 | 3.36 | 3.16 | 14336 | 17.03 | 23.80 | 20.42 |
| 14. | WH 1080 | 1.41 | 2.24 | 1.83 | PI 604173 | 12.61 | 17.54 | 15.08 |
| 15. | PBW 550 | 13.01 | 14.79 | 13.90 | PI 604172 | 22.39 | 27.63 | 25.01 |
| 16. | PBW 373 | 12.20 | 13.59 | 12.89 | 15 | 15.42 | 20.64 | 18.03 |
| 17. | DBW 88 | 8.97 | 9.80 | 9.38 | 13762 | 19.37 | 25.61 | 22.49 |
| 18. | WH 1124 | 5.44 | 5.58 | 5.51 | 54 | 18.20 | 23.76 | 20.98 |
| 19. | PBW 396 | 7.49 | 7.99 | 7.74 | PI 604162 | 13.36 | 19.34 | 16.35 |
| 20. | PBW 644 | 12.30 | 13.62 | 12.96 | PI 604185 | 19.98 | 25.49 | 22.73 |
| 21. | WH 1142 | 2.60 | 2.82 | 2.71 | 13765 | 15.18 | 16.32 | 15.75 |
| 22. | PBW 343 | 15.44 | 16.62 | 16.03 | 9798 | 14.22 | 17.27 | 15.74 |
| 23. | HD 3086 | 13.43 | 13.76 | 13.59 | PI 604186 | 12.09 | 14.91 | 13.50 |
| 24. | RAJ 3077 | 4.98 | 5.37 | 5.17 | | | | |
| 25. | WH 1105 | 13.61 | 15.13 | 14.37 | | | | |
| | C.D. (p = 0.05) | 0.445 | 0.327 | | | NS | 9.678 | |
| | SE(d) | 0.220 | 0.162 | | | 4.673 | 4.786 | |
| | C.V. | 3.014 | 2.058 | | | 34.284 | 27.431 | |

This was followed by markers *Xgwm337*, *Xgwm136*, and *Xgwm77* with MAF values of 0.903. On the other hand, the marker *Xgwm113* had the lowest MAF of 0.552, indicating a relatively lower frequency of the most common allele at this marker locus. This was followed by marker *Xgwm113* (0.552), *Xgwm301* (0.579), *Xpsp3029* and *Xgwm383* (0.583) (Table 4).

The gene diversity (GD) values for the studied SSR markers ranged from 0.115 to 0.489, indicating the level of genetic diversity present at each marker locus within the studied wheat genotypes.

The gene diversity (GD) values for the examined SSR markers varied from 0.115 to 0.489 demonstrating the degree of genetic variation present at each marker locus within the studied wheat genotypes. The average GD across all markers was 0.351 was recording, pointing a modest degree of genetic variety within the population moderate level of genetic diversity within the population. The marker *Xgwm113* had the highest GD value of 0.489, demonstrating a relatively high level of genetic diversity at this locus. This was followed by markers *Xpsp3029* with a GD of 0.483, *Xgwm301* and *Xgwm540* with GD values of 0.479 and 0.475, respectively. On the other hand, the marker *Xgwm526* had the lowest GD value of 0.115, suggesting a relatively lower level of genetic diversity at this locus. This was followed by markers *Xcfd45*, *Xgwm337* and *Xgwm136* with GD values of 0.117, 0.170 and 0.173, respectively (Table 4).

**Table 4. Major allele frequency (MAF), gene diversity (GD) and polymorphic information content (PIC) values of 51 polymorphic microsatellite markers.**

| Sr. No. | Marker | Major Allele Frequency (MAF) | Gene Diversity (GD) | Polymorphic Information Content (PIC) |
|---|---|---|---|---|
| 1. | Xpsp3000 | 0.715 | 0.360 | 0.284 |
| 2. | Xgwm44 | 0.667 | 0.402 | 0.315 |
| 3. | Xpsp3079 | 0.802 | 0.300 | 0.249 |
| 4. | Xgwm111 | 0.823 | 0.254 | 0.208 |
| 5. | Xgwm106 | 0.604 | 0.464 | 0.356 |
| 6. | Xgwm337 | 0.903 | 0.170 | 0.151 |
| 7. | Xgwm642 | 0.729 | 0.339 | 0.270 |
| 8. | Xgwm136 | 0.903 | 0.173 | 0.157 |
| 9. | Xcfd45 | 0.938 | 0.117 | 0.110 |
| 10. | Xgwm473 | 0.760 | 0.364 | 0.298 |
| 11. | Xgwm635 | 0.688 | 0.430 | 0.337 |
| 12. | Xgwm174 | 0.783 | 0.303 | 0.249 |
| 13. | Xpsp3029 | 0.583 | 0.483 | 0.366 |
| 14. | Xgwm260 | 0.594 | 0.465 | 0.356 |
| 15. | Xgwm121 | 0.760 | 0.355 | 0.290 |
| 16. | Xgwm148 | 0.736 | 0.360 | 0.289 |
| 17. | Xgwm495 | 0.885 | 0.201 | 0.180 |
| 18. | Xgwm265 | 0.818 | 0.285 | 0.239 |
| 19. | Xgwm2 | 0.642 | 0.432 | 0.333 |
| 20. | Xgwm391 | 0.792 | 0.312 | 0.256 |
| 21. | Xgwm165 | 0.674 | 0.436 | 0.341 |
| 22. | Xgwm637 | 0.854 | 0.205 | 0.168 |
| 23. | Xgwm304 | 0.854 | 0.248 | 0.217 |
| 24. | Xgwm276 | 0.729 | 0.387 | 0.311 |
| 25. | Xgwm153 | 0.646 | 0.413 | 0.321 |
| 26. | Xgwm526 | 0.938 | 0.115 | 0.107 |
| 27. | Xgwm77 | 0.903 | 0.163 | 0.142 |
| 28. | Xgwm299 | 0.625 | 0.458 | 0.352 |
| 29. | Xgwm113 | 0.552 | 0.489 | 0.369 |
| 30. | Xgwm537 | 0.708 | 0.410 | 0.325 |
| 31. | Xgwm540 | 0.604 | 0.475 | 0.362 |
| 32. | Xgwm335 | 0.650 | 0.443 | 0.344 |
| 33. | Xgwm508 | 0.681 | 0.414 | 0.325 |
| 34. | Xgwm193 | 0.667 | 0.413 | 0.324 |
| 35. | Xgwm146 | 0.833 | 0.225 | 0.184 |
| 36. | Xgwm210 | 0.708 | 0.366 | 0.291 |
| 37. | Xgwm301 | 0.579 | 0.479 | 0.364 |
| 38. | Xgwm314 | 0.615 | 0.463 | 0.355 |
| 39. | Xgwm383 | 0.583 | 0.472 | 0.360 |
| 40. | Xgwm194 | 0.938 | 0.117 | 0.110 |
| 41. | Xgwm192 | 0.778 | 0.339 | 0.279 |
| 42. | Xcfd14 | 0.646 | 0.446 | 0.346 |
| 43. | Xcfd68 | 0.802 | 0.269 | 0.215 |
| 44. | Xpsp3200 | 0.667 | 0.444 | 0.346 |
| 45. | Xbarc128 | 0.681 | 0.409 | 0.323 |
| 46. | Xbarc148 | 0.764 | 0.328 | 0.265 |
| 47. | Xbarc171 | 0.792 | 0.322 | 0.268 |

*(Continued)*

**Table 4.** (Continued)

| Sr. No. | Marker | Major Allele Frequency (MAF) | Gene Diversity (GD) | Polymorphic Information Content (PIC) |
|---|---|---|---|---|
| 48. | *Xbarc172* | 0.708 | 0.391 | 0.311 |
| 49. | *Xbarc17* | 0.609 | 0.449 | 0.345 |
| 50. | *Xbarc126* | 0.688 | 0.400 | 0.317 |
| 51. | *Xbarc214* | 0.771 | 0.343 | 0.281 |
| **Mean** | | **0.733** | **0.351** | **0.280** |

The SSR marker's computed polymorphic information content (PIC) values were found in the range from 0.107 to 0.369, indicating the level of informativeness of each marker in capturing genetic variation within the studied wheat genotypes. The average PIC value across all markers was 0.280, indicating that the population's markers are only moderately informative. The marker *Xgwm113* had the highest PIC value of 0.369, indicating its high level of informativeness in capturing genetic diversity. This was followed by markers *Xpsp3029*, *Xgwm301* and *Xgwm383* having 0.366, 0.364 and 0.360 PIC values, respectively. However, the marker *Xgwm526* had the lowest PIC value of 0.107, suggesting its lower informativeness in capturing genetic diversity. This was followed by markers *Xcfd45* and *Xgwm194*, both with a PIC of 0.110, and then by markers *Xgwm77* and *Xgwm337* having 0.142 and 0.151 PIC values, respectively (Table 4).

## Population structure analysis

In order to investigate the population structure and identify subgroups within the studies wheat genotypes, the data obtained from the 51 SSR markers were analysed using STRUCTURE analysis. The admixture model was used, and a threshold of 60 per cent membership probability was employed to allocate genotypes to a particular cluster. Based on ΔK values for K ranging from 3 to 9 (Fig 1), the STRUCTURE analysis findings showed the occurrence of four major sub-groups within the populations (Fig 2). These sub-groups were represented by different colors: red cluster (4 genotypes), green cluster (20 genotypes), yellow cluster (8 genotypes), and blue cluster (14 genotypes). The number of genotypes within each sub-group suggested varying levels of genetic differentiation among the studied wheat genotypes. Two genotypes, G13 and W91 (UP 2425 and wild accession 13993), failed to achieve the threshold membership probability for inclusion under any of the clusters. These genotypes were considered as admixtures, indicating that they possess genetic characteristics from multiple sub-groups. The findings of the population structure analysis were further supported by the construction of a UPGMA-based phylogenetic tree using DARwin 6 software (Fig 3). The genetic linkages and clustering patterns within the wheat genotypes were recorded visually by the phylogenetic tree. Additionally, principal coordinate analysis (PCoA) was performed by employing software Past 4.08 to further explore the population structure (Fig 4). This multivariate technique helped in visualization of genetic distances and patterns of differentiation among individuals or groups. The PCoA results likely confirmed the presence of the identified sub-groups and provided additional insights into the genetic makeup of the genotypes of wheat under investigation.

Overall, detailed information on the population structure and genetic relationships among the wheat genotypes under study was obtained by the combination of STRUCTURE analysis, phylogenetic tree building, and PCoA.

## Analysis of Molecular Variance (AMOVA)

Analysis of Molecular Variance (AMOVA) was conducted using GenAlEx version 6.5 to assess the partitioning of genetic variation within the wheat germplasm. The findings showed that

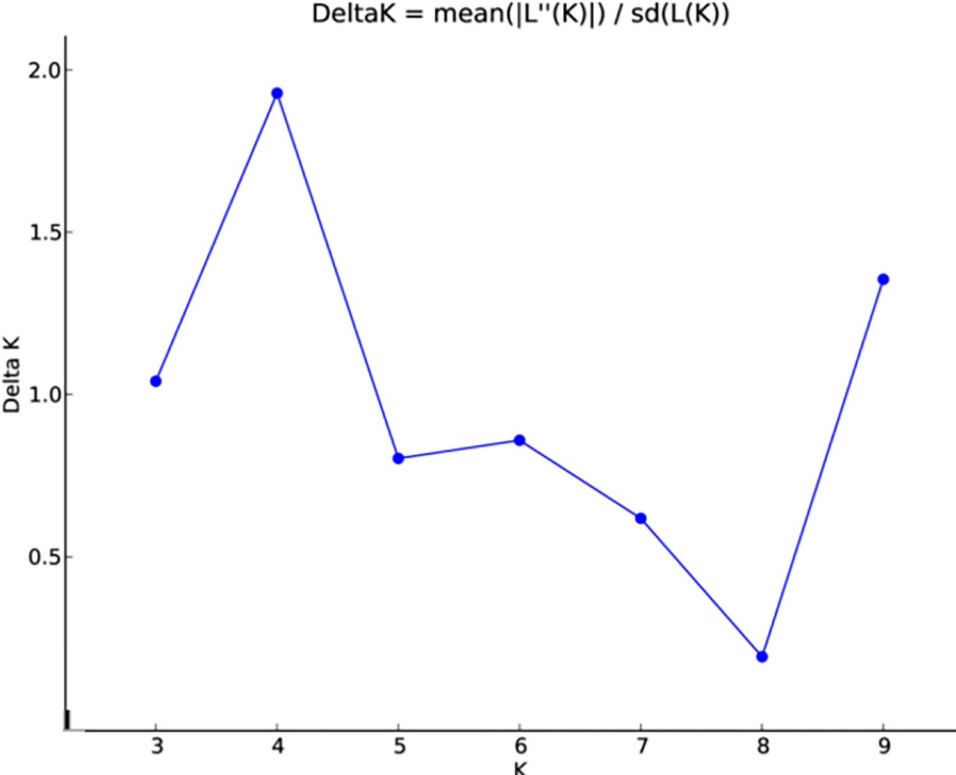

**Fig 1. Estimation of number of clusters using ΔK values for K ranging from 3 to 9.**

the distinction between bread wheat and wild wheat genotypes accounted for around 37 per cent of the overall variance. However, 63 per cent of the variation was found within individuals within each group, indicating a substantial level of diversity within both bread wheat and wild wheat genotypes (Table 5). Regarding the genetic diversity parameters, the number of different alleles per locus (Na) was found to be 1.608 and 1.776 for bread wheat and wild wheat genotypes, respectively, with an average value of 1.692. The number of effective alleles per locus (Ne) was calculated as 1.384 and 1.521 for bread wheat and wild wheat genotypes, respectively, with a mean value of 1.453 (Table 6).

For bread wheat genotypes and wild wheat germplasm, the Shannon's Information Index (I), which gauges population genetic diversity, showed mean values of 0.362 and 0.452, respectively. These values revealed that the wild wheat genotypes have more genetic diversity than the bread wheat genotypes (Table 6).

With an average of 0.268 for the whole wheat germplasm, the gene diversity (h) value, which calculates the likelihood that two randomly chosen alleles are distinct within a

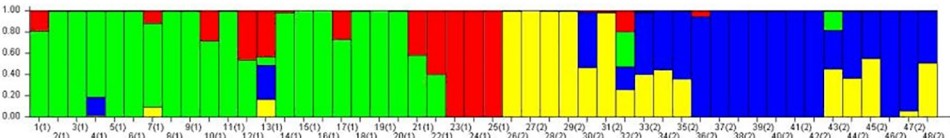

**Fig 2. Bar graph for population structure of selected wheat genotypes performed by admixture method in STRUCTURE.**

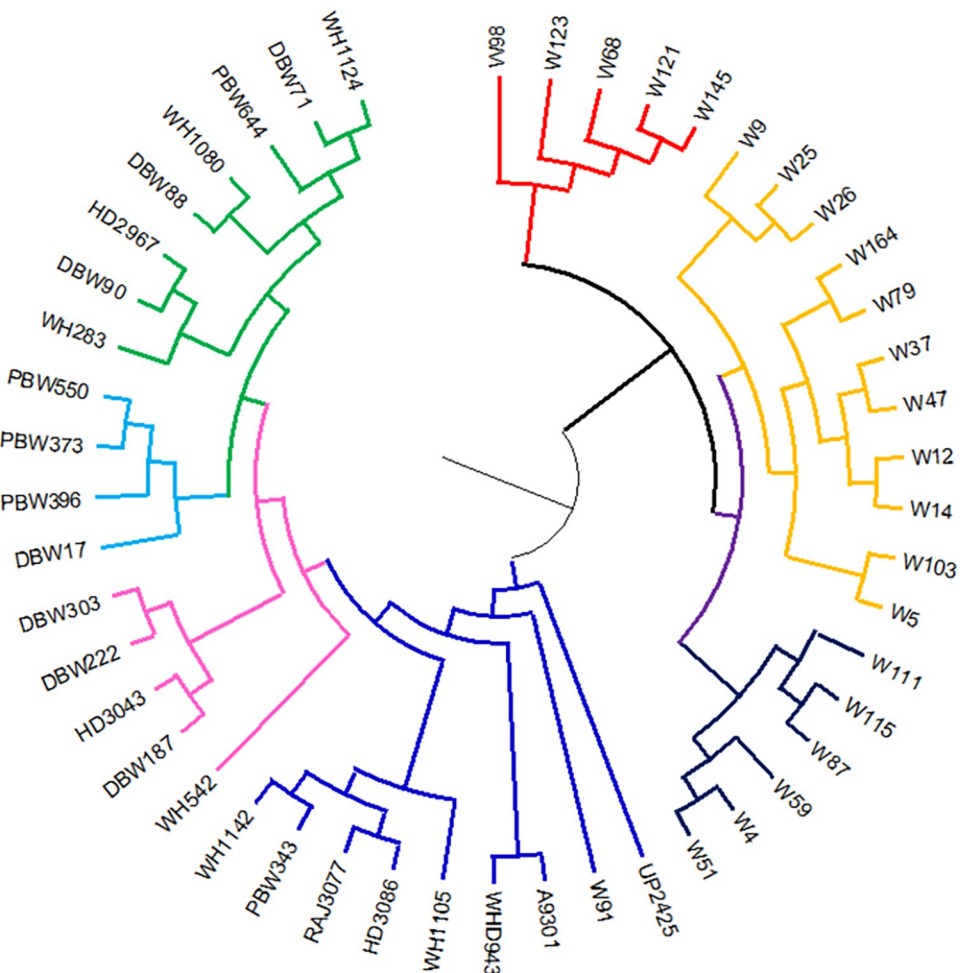

**Fig 3. Phylogenetic tree construction for selected bread wheat and wild wheat genotypes using UPGMA based clustering method.**

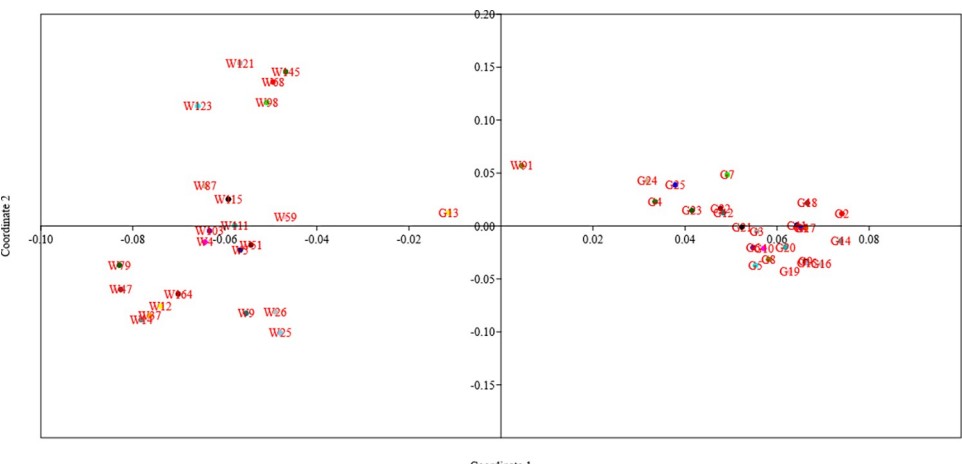

**Fig 4. Principal coordinate analysis (PCoA) for selected bread wheat and wild wheat genotypes for population structure analysis.**

**Table 5. Analysis of Molecular Variance (AMOVA) of selected bread wheat and wild wheat genotypes.**

| Source | Degree of freedom (Df) | Sum of square (SS) | Mean sum of square(MS) | Estimated variance | Percent of variance |
|---|---|---|---|---|---|
| Among populations | 1 | 297.038 | 297.038 | 11.567 | 37 |
| Within populations | 46 | 916.275 | 19.919 | 19.919 | 63 |
| Total | 47 | 1213.313 | | 31.486 | 100 |

population, was 0.235 for bread wheat genotypes and 0.302 for wild wheat genotypes. The difference between the unbiased diversity (uh) and gene diversity (h) values was marginal. Specifically, the uh value was 0.245 for bread wheat genotypes and 0.316 for wild wheat genotypes, with an average of 0.280 (Table 6).

## Linkage Disequilibrium (LD) and association mapping

Utilising 51 SSR markers from the wheat genotype panel, a total of 9453 locus pairs were analysed for linkage disequilibrium (LD). Amongst the observed locus pairs, 3546 pairs (36.56 per cent) had significant LD at the threshold value of $r^2 \geq 0.05$, revealing a non-random association between alleles at these loci. When a higher threshold of $r^2 \geq 0.1$ was applied, significant LD was recorded in 2029 marker pairs (21.46 per cent).

Furthermore, out of the total locus pairs, 1020 pairs (10.79 per cent) exhibited significant LD at a p-value of $\leq 0.005$, while 1256 pairs showed significant LD at a p-value of $\leq 0.01$. Additionally, 2040 locus pairs (21.58 per cent) displayed a D' value of 1, indicating complete linkage or co-inheritance of these loci (Figs 5 and 6). In terms of Marker-Trait Associations (MTAs) related to aphid resistance, 23 significant MTAs were identified during the 2018–19 cropping season at a p-value of $\leq 0.009$. Among these, the microsatellite marker *Xpsp3029* showed the highest $r^2$ value of 0.7240, followed by *Xgwm44* (0.5898), *Xgwm174* (0.4387), and *Xgwm301* (0.4087). In the subsequent cropping season (2019–20), 27 significant MTAs were recorded at a p-value of $\leq 0.008$. The marker *Xgwm44* had the highest $r^2$ value of 0.9646, followed by *Xpsp3029* (0.9387), *Xgwm2* (0.7589), and *Xgwm121* (0.7109). Among the significant MTAs, 22 were found to be common between the two cropping seasons (2018–19 and 2019–20) (Table 7).

## Discussion

In the present study, the experimental material consisted of 48 wheat genotypes, which included 25 bread wheat genotypes belonging to the *Triticum* spp. and 23 wild wheat genotypes belonging to *Aegilops* spp. & *T. dicoccoides*. These genotypes were selected for their

**Table 6. Genetic diversity and mean allelic pattern across selected bread wheat and wild wheat genotypes.**

| Population | Released Varieties | | Wild Accessions | | Total Mean |
|---|---|---|---|---|---|
| | Mean | ±SE | Mean | ±SE | |
| N | 25.000 | 0.000 | 23.000 | 0.000 | 24.000 |
| Na | 1.608 | 0.064 | 1.776 | 0.052 | 1.692 |
| Ne | 1.384 | 0.028 | 1.521 | 0.030 | 1.453 |
| I | 0.362 | 0.021 | 0.452 | 0.020 | 0.407 |
| h | 0.235 | 0.015 | 0.302 | 0.015 | 0.268 |
| uh | 0.245 | 0.016 | 0.316 | 0.015 | 0.280 |

**Na** = No. of Different Alleles; **Ne** = No. of Effective Alleles = $1 / (p^2 + q^2)$; **I** = Shannon's Information Index = $-1 * (p * Ln(p) + q * Ln(q))$; **h** = Diversity = $1 - (p^2 + q^2)$; **uh** = Unbiased Diversity = $(N / (N-1)) * h$; Where for Haploid Binary data, p = Band Freq. and q = 1—p.

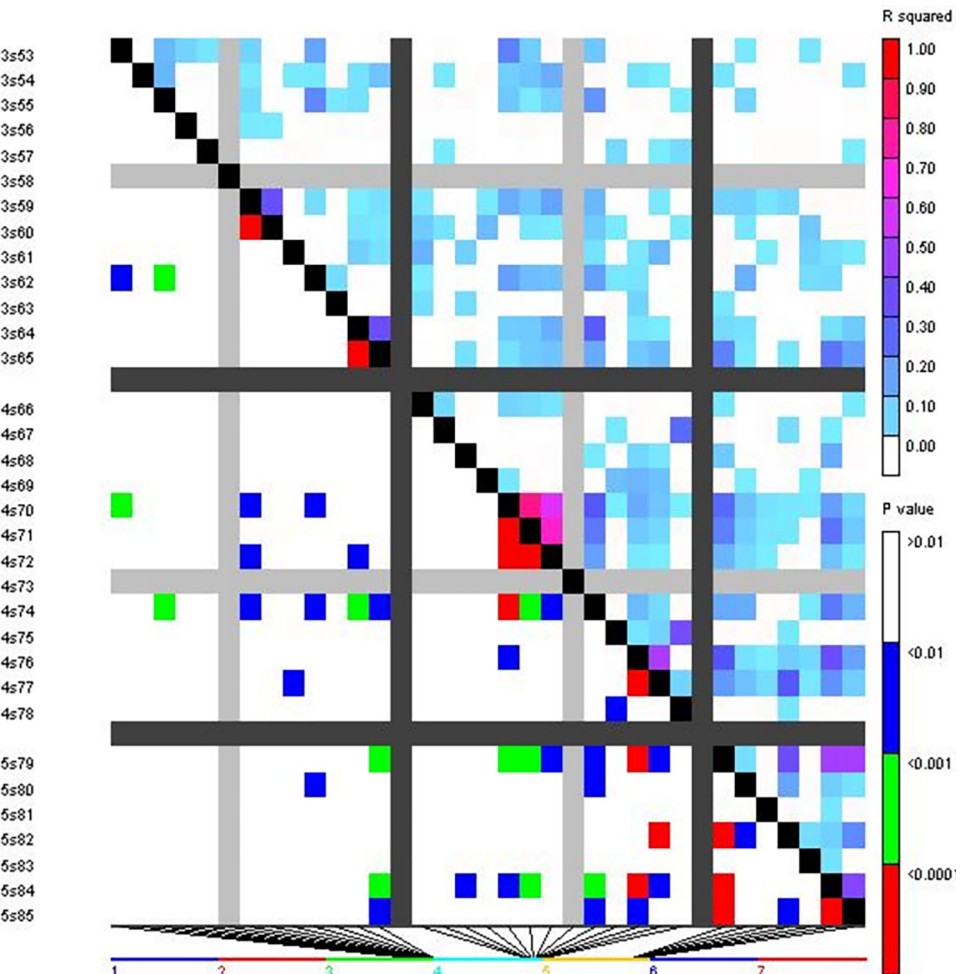

**Fig 5. Triangle heat plot showing pairwise values of $r^2$ and $p$ for different locus pairs over different chromosomes of selected wheat genotypes.**

relevance to the investigation of *R. maidis* resistance. There were 51 polymorphic markers employed in total for the investigation of molecular diversity. With an average of 7.29 markers per chromosome, these markers were distributed across 7 chromosomes. The valuable information was obtained about the genetic variation present in the studied wheat genotypes by studying markers. With an average of 2.804 alleles per marker, the 51 polymorphic markers showed a range of 2 to 5 alleles per marker. The average values for major allele frequency, gene diversity, and polymorphic information content were reported as 0.733, 0.351, and 0.280, respectively. These MAF values provide insights into the allelic frequencies and the distribution of major alleles at each marker locus within the studied wheat genotypes. These results in the current investigation suggested that the wheat genotypes under examination exhibited a modest amount of genetic variety. This range of alleles shows that the genetic diversity captured by the markers and allows for further analysis of allelic variation and its relationship with *R. maidis* resistance. The combination of the selected wheat genotypes and the polymorphic markers provided a comprehensive set of materials for the molecular categorization of *R. maidis* resistance and the evaluation of genetic diversity within the studied wheat population.

Comparing these findings with previous studies, Peng et al. [58] analyzed 71 wheat accessions using 51 SSR markers and reported higher diversity values, with a mean number of SSR

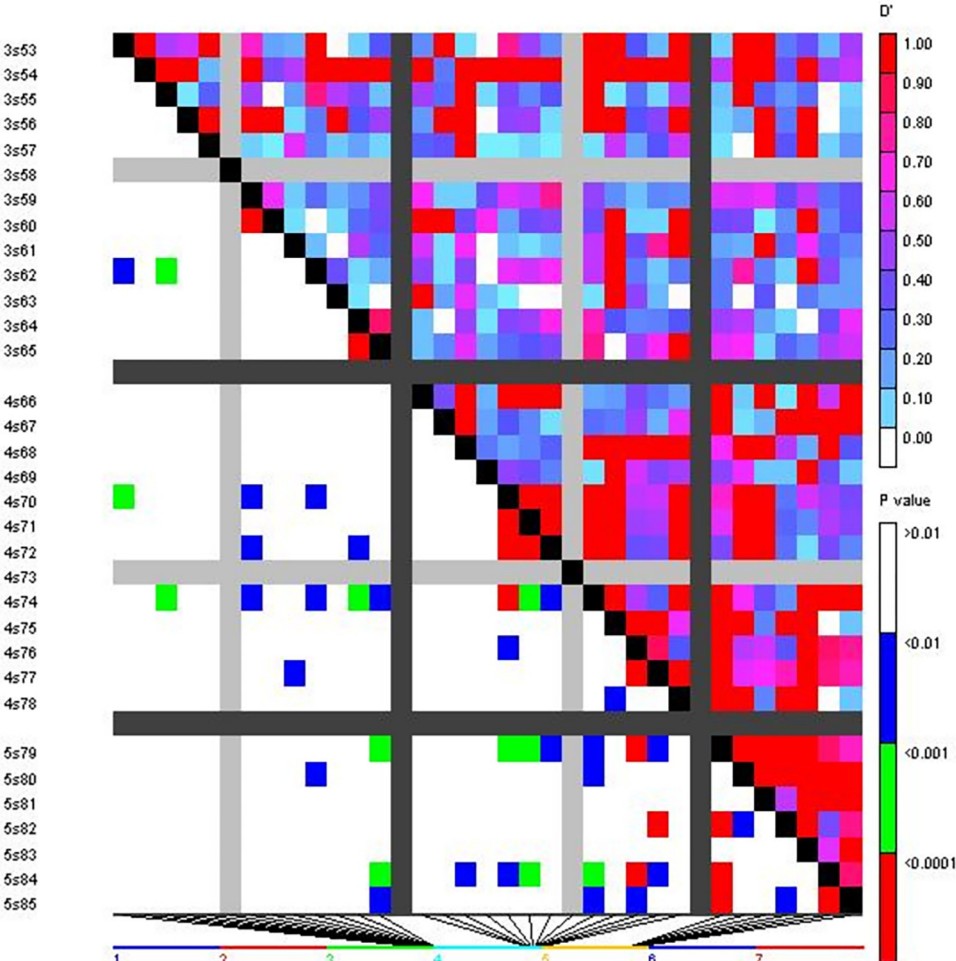

**Fig 6. Triangle heat plot showing pairwise values of *D'* and *p* for different locus pair over different chromosomes of selected wheat genotypes.**

alleles per locus of 6.7, mean Shannon's index of 1.291, and mean Nei's gene diversity of 0.609. Similarly, Li and Peng [59] evaluated 70 bread wheat accessions using the same 51 SSR markers and found 593 alleles and 97 polymorphic loci, with an average of 6.11 polymorphic alleles and 3.38 effective alleles. The mean values for polymorphic allele number, effective allele number, Shannon's information index, Nei's gene diversity, and genetic distance were 4.44, 3.10, 1.11, 0.57, and 0.89, respectively. Liu et al. [60] conducted a study using 99 wheat SSR markers and found an average of 10.48 alleles per locus, ranging from 2 to 30 alleles. The polymorphism information content (PIC) values ranged from 0.178 to 0.973, with an average of 0.717. These comparisons indicate that the genetic diversity observed in the present study is slightly lower than some of the previous studies, suggesting potential variations in the diversity levels among different sets of wheat genotypes and marker systems used.

The STRUCTURE analysis identified four major sub-groups within studied wheat germplasm based on ΔK values for K ranging from 3 to 9. These sub-groups were represented by different colors, namely the Red cluster (containing 4 genotypes), Green cluster (containing 20 genotypes), Yellow cluster (containing 8 genotypes), and Blue cluster (containing 14 genotypes). Two genotypes, UP 2425 and wild genotype 13993, were regarded as admixtures since

**Table 7. Significant marker-trait associations (MTAs) during 2018–19 and 2019–20 cropping seasons using Generalised Mixed Model (GLM).**

| Sr. No. | 2018–19 | | | | 2019–20 | | | |
|---|---|---|---|---|---|---|---|---|
| | SSR | Chromosome | $p$ value | $r^2$ | Marker | Chromosome | $p$ value | $r^2$ |
| 1. | Xbarc17 | 1 | 0.004 | 0.1655 | Xgwm106 | 1 | 0.002 | 0.1954 |
| 2. | Xgwm106 | 1 | 0.002 | 0.1902 | Xgwm153 | 1 | 0.002 | 0.1979 |
| 3. | Xbarc148 | 1 | 0.001 | 0.2147 | Xbarc17 | 1 | 0.000 | 0.2772 |
| 4. | Xgwm301 | 2 | 0.000 | 0.4087 | Xbarc148 | 1 | 0.000 | 0.2199 |
| 5. | Xbarc128 | 2 | 0.001 | 0.2054 | Xbarc128 | 2 | 0.001 | 0.4837 |
| 6. | Xpsp3029 | 2 | 0.000 | 0.7240 | Xgwm301 | 2 | 0.000 | 0.4917 |
| 7. | Xgwm391 | 3 | 0.007 | 0.1478 | Xpsp3029 | 2 | 0.000 | 0.9387 |
| 8. | Xgwm299 | 3 | 0.005 | 0.1608 | Xgwm299 | 3 | 0.008 | 0.1426 |
| 9. | Xgwm314 | 3 | 0.001 | 0.2287 | Xgwm391 | 3 | 0.004 | 0.1685 |
| 10. | Xgwm2 | 3 | 0.000 | 0.2510 | Xgwm2 | 3 | 0.001 | 0.7589 |
| 11. | Xgwm113 | 4 | 0.009 | 0.1406 | Xgwm314 | 3 | 0.00 | 0.2458 |
| 12. | Xgwm165 | 4 | 0.005 | 0.3152 | Xgwm113 | 4 | 0.001 | 0.4094 |
| 13. | Xgwm637 | 4 | 0.000 | 0.3890 | Xgwm165 | 4 | 0.004 | 0.6071 |
| 14. | Xgwm335 | 5 | 0.006 | 0.3137 | Xgwm637 | 4 | 0.000 | 0.4515 |
| 15. | Xgwm174 | 5 | 0.004 | 0.4387 | Xgwm121 | 5 | 0.004 | 0.7109 |
| 16. | Xgwm121 | 5 | 0.000 | 0.3062 | Xgwm192 | 5 | 0.006 | 0.3013 |
| 17. | Xgwm193 | 6 | 0.004 | 0.1668 | Xgwm335 | 5 | 0.004 | 0.5436 |
| 18. | Xbarc214 | 7 | 0.007 | 0.3001 | Xgwm174 | 5 | 0.001 | 0.4818 |
| 19. | Xgwm44 | 7 | 0.000 | 0.5898 | Xgwm508 | 6 | 0.006 | 0.4529 |
| 20. | Xgwm121 | 7 | 0.007 | 0.1486 | Xbarc126 | 7 | 0.008 | 0.1446 |
| 21. | Xgwm276 | 7 | 0.003 | 0.1811 | Xbarc171 | 6 | 0.002 | 0.3730 |
| 22. | Xcfd14 | 7 | 0.001 | 0.2216 | Xgwm193 | 6 | 0.000 | 0.2722 |
| 23. | Xgwm260 | 7 | 0.001 | 0.2197 | Xgwm260 | 7 | 0.005 | 0.4624 |
| 24. | | | | | Xgwm44 | 7 | 0.000 | 0.9646 |
| 25. | | | | | Xgwm276 | 7 | 0.000 | 0.2232 |
| 26. | | | | | Xbarc214 | 7 | 0.000 | 0.4749 |
| 27. | | | | | Xcfd14 | 7 | 0.00 | 0.3121 |

they failed to meet the membership probability criterion for any of the clusters. Comparing these findings with the study by Li and Peng [59], two distinct groups were identified in the wheat germplasm using the STRUCTURE Bayesian analysis, with the highest ΔK value obtained at 2 suggesting the wheat germplasm could be divided into two groups. Following a population structure study, Kisten et al. [35] separated the wheat panel into two unique clusters. Cluster 2 featured 67 bulked samples of spring-type breeding lines created in Montana, United States, whereas Cluster 1 contained 102 bulked samples of winter-type cultivars and breeding lines from the United States and Turkey.

These results from different studies demonstrate the presence of population sub-structuring in wheat germplasm, with varying numbers of clusters identified. The clustering patterns may vary depending on the specific set of genotypes, geographical origin, breeding history, and other factors influencing the genetic makeup of the wheat germplasm under investigation.

Pairwise analysis performed to determine the level of Linkage Disequilibrium (LD) in wheat genotype panel, resulted in the detection of LD for 9453 locus pairs. Out of the total 9453 locus pairs analyzed, 3546 locus pairs (approximately 36.56 per cent) demonstrated a significant LD (threshold: $r^2 \geq 0.05$. This indicates a non-random co-segregation of alleles at these loci. When a higher threshold of $r^2 \geq 0.1$ was applied, significant LD was observed for 2029 marker pairs, accounting for approximately 21.46 per cent of the total locus pairs

analyzed. This suggested a stronger linkage between these markers. Additionally, 2040 locus pairs (approximately 21.58 per cent) displayed a D' value of 1, indicating complete linkage between these loci. This implies that these locus pairs are co-inherited and are in strong genetic linkage. The findings provide insights into the extent and patterns of LD within the wheat genotype panel, highlighting the presence of significant LD and co-inherited loci. These results have implications for understanding the genetic architecture and potential marker-trait associations within the wheat population under investigation.

Marker-Trait Associations (MTAs) was studied using a Generalized Mixed Model (GLM) with a threshold of -log10(p-value) $\leq$ 2 applied to declare significant MTAs. During the cropping season of 2018–19, a total of 23 significant MTAs associated with aphid resistance were observed at a significance level of p $\leq$ 0.009. Similarly, in the following cropping season of 2019–20, 27 significant MTAs were found at a significance level of p $\leq$ 0.008. Importantly, 22 of these significant MTAs were found to be common between the two cropping seasons, suggesting consistent associations with aphid resistance.

Association mapping studies in wheat for detecting aphid resistance are relatively limited. Previous genetic studies primarily focused on linkage mapping using bi-parental mapping populations, which have lower mapping resolution compared to association mapping. Association mapping utilizes historical recombination events and benefits from higher allelic diversity [61]. The mapping investigations use the cumulative recombination events across generations to find relationships between genotypes and phenotypes based on linkage disequilibrium [62]. Some loci associated with aphid resistance have been found to be shared across different species of aphids, potentially indicating their location in resistance gene clusters within specific chromosomal regions of wheat. For instance, *Xgwm111*, which is linked to other resistance-related genes, has been reported to be associated with aphid resistance [63, 64]. Previous association mapping studies by Peng et al. [58], Dahleen et al. [65], Li and Peng [59], and Kisten et al. [35] have also identified significant associations between SSR loci or SNP markers and aphid resistance or related traits in wheat germplasm. The results obtained from association mapping provide valuable insights into the genetic basis of aphid resistance mechanism in wheat, contributing towards understanding of the underlying loci and potential markers associated with resistance traits.

## Conclusion

The extensive phenotyping conducted on the wheat genotypes revealed significant variation in terms of aphid resistance. Molecular analysis using polymorphic markers further divided the genotypes into four clusters, indicating the presence of evolutionary relationships among bread wheat genotypes and wild accessions or relatives of wheat. This suggests that wild accessions or landraces of wheat could potentially serve as sources for aphid resistance. LD (linkage disequilibrium) based association mapping was performed on the 48 genotypes using 51 polymorphic markers. It was observed that 21.58 per cent of the locus pairs exhibited a D' value of 1, indicating co-inheritance of these markers. Furthermore, 22 significant marker-trait associations (MTAs) were identified in both cropping seasons, highlighting their consistency across different years. However, genetic analysis of aphid resistance in wheat poses challenges due to uncontrollable environmental factors. It is difficult to maintain consistent levels of aphid infestation across segregating generations of breeding material, leading to the possibility of susceptible plants escaping selection. However, the study's findings help to elucidate the genetic factors behind aphid resistance. Molecular markers assisted selection (MAS) is recommended as the optimal approach for selecting wheat plants with aphid tolerance during breeding programs and introducing resistance into susceptible genotypes. By identifying the morphological

and molecular markers connected to aphid resistance and associating these markers with genes can be used in wheat resistance breeding programmes. The results emphasize the importance of genetic diversity in the studied wheat genotypes and provide insights into population structure, genetic relationships, and marker-trait associations, particularly for traits such as aphid resistance.

## Acknowledgments

The authors are thankful to the Director, ICAR-Indian Institute of Wheat and Barley Research, Karnal for providing the necessary research facilities for the study. The authors also acknowledge the support provided by the technical staff of Crop Protection Division of the institute.

## Author Contributions

**Conceptualization:** Jayant Yadav, Poonam Jasrotia, Maha Singh Jaglan, Sindhu Sareen, Surender Singh Yadav, Gyanendra Singh, Gyanendra Pratap Singh.

**Data curation:** Jayant Yadav, Poonam Jasrotia.

**Formal analysis:** Jayant Yadav, Poonam Jasrotia, Maha Singh Jaglan, Prem Lal Kashyap, Sudheer Kumar, Surender Singh Yadav.

**Funding acquisition:** Gyanendra Singh.

**Investigation:** Poonam Jasrotia, Prem Lal Kashyap, Gyanendra Pratap Singh.

**Methodology:** Poonam Jasrotia, Sindhu Sareen, Prem Lal Kashyap.

**Project administration:** Poonam Jasrotia, Maha Singh Jaglan, Sudheer Kumar, Gyanendra Singh, Gyanendra Pratap Singh.

**Resources:** Sindhu Sareen, Prem Lal Kashyap.

**Supervision:** Poonam Jasrotia, Maha Singh Jaglan, Sindhu Sareen, Prem Lal Kashyap, Sudheer Kumar, Surender Singh Yadav, Gyanendra Singh, Gyanendra Pratap Singh.

**Writing – original draft:** Jayant Yadav, Poonam Jasrotia.

**Writing – review & editing:** Poonam Jasrotia, Sindhu Sareen, Prem Lal Kashyap, Sudheer Kumar, Surender Singh Yadav, Gyanendra Singh, Gyanendra Pratap Singh.

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
