## [Decision Letter · Decision Letter 0]

19 Sep 2023

PONE-D-23-22687Unravelling the novel genetic diversity and marker-trait associations of corn leaf aphidPLOS ONE

Dear Dr. Jasrotia,

Thank you for submitting your manuscript to PLOS ONE. After careful consideration, we feel that it has merit but does not fully meet PLOS ONE’s publication criteria as it currently stands. Therefore, we invite you to submit a revised version of the manuscript that addresses the points raised during the review process.

We look forward to receiving your revised manuscript.

Kind regards,

Mehdi Rahimi, Ph.D.

Academic Editor

PLOS ONE

Journal Requirements:

   "Yes. The funding for conducting the experiment was provided under  the Institute project “Management of wheat insect-pests through climate-smart pest management strategies” of ICAR-ICAR- Indian Institute of Wheat and Barley Research, Karnal 132001, Haryana, India."

7. PLOS requires an ORCID iD for the corresponding author in Editorial Manager on papers submitted after December 6th, 2016. Please ensure that you have an ORCID iD and that it is validated in Editorial Manager. To do this, go to ‘Update my Information’ (in the upper left-hand corner of the main menu), and click on the Fetch/Validate link next to the ORCID field. This will take you to the ORCID site and allow you to create a new iD or authenticate a pre-existing iD in Editorial Manager. Please see the following video for instructions on linking an ORCID iD to your Editorial Manager account: https://www.youtube.com/watch?v=_xcclfuvtxQ

8. Please amend either the title on the online submission form (via Edit Submission) or the title in the manuscript so that they are identical.

9. Please include your full ethics statement in the ‘Methods’ section of your manuscript file. In your statement, please include the full name of the IRB or ethics committee who approved or waived your study, as well as whether or not you obtained informed written or verbal consent. If consent was waived for your study, please include this information in your statement as well. 

Reviewers' comments:

Reviewer's Responses to Questions

**Comments to the Author**

1. Is the manuscript technically sound, and do the data support the conclusions?

Reviewer #1: Yes

Reviewer #2: Yes

2. Has the statistical analysis been performed appropriately and rigorously? 

Reviewer #1: Yes

Reviewer #2: Yes

3. Have the authors made all data underlying the findings in their manuscript fully available?

Reviewer #1: No

Reviewer #2: Yes

4. Is the manuscript presented in an intelligible fashion and written in standard English?

Reviewer #1: Yes

Reviewer #2: Yes

5. Review Comments to the Author

Reviewer #1: The manuscript entitled "Unravelling the novel genetic diversity and marker-trait associations of corn leaf aphid" is dealing with one of the most important insect of wheat. The study reports important marker trait associations genes for corn leaf aphid. The study is important and I recommend acceptance of this manuscript after the authors make following changes in the manuscript:

1. A few other aphid species also infest wheat crop in north-western plains of Punjab and Haryana. So, how you ensure that the standing wheat crop was only infested with R. maidis and no other aphid species were present?

2. In the whole manuscript the ‘%’ sign should be replaced with ‘per cent’ except those in the brackets.

3. On Page no. 4, Line no. 86, the sentence ‘DNA markers linked to insect resistance genes can be identified using Simple-Sequence Repeats (SSRs)’ needs to be revised because SSRs are itself DNA markers.

4. On Page no. 4, Line no. 91, insert space between ‘breeding process’ and ‘(Wani et al. 2022)’.

5. What was the criteria for selection and why only these 48 wheat germplasms were taken into consideration?

6. At which time or stage or interval, the number of aphids infesting wheat plants was counted for phenotyping.

7. On Page no. 10, Line no. 219, insert space between ‘-log10(p-value) ≤ 2’ and ‘To visualize’.

8. As per the manuscript, to visualize the results of association mapping, Manhattan plots were created. But why these are not included in the manuscript?

9. On Page no. 19, Line no. 442, insert space between ‘aphid resistance.’ and ‘LD’.

Reviewer #2: Overall, the research paper titled " Unravelling the novel genetic diversity and marker-trait associations of corn leaf aphid" explores an interesting and relevant topic related to corn leaf aphids in wheat. The authors have conducted a comprehensive study on this pest's genetic diversity and marker-trait associations. However, there are several areas that need improvement and clarification before the paper can be considered for publication:

1. The title should give a good overview of the paper's focus. The present title is unclear and misleading about whether the work was done using corn or wheat. Authors must include the crop name in the title for clarity of the work.

2. The use of SSR markers is appropriate for studying genetic diversity, but it would be beneficial to discuss why SSRs were chosen over other marker types (e.g., SNP markers) and their advantages and limitations in this context.

3. Ensure that the references cited in the manuscript are up-to-date and relevant to the topic and also correctly formatted.

6. PLOS authors have the option to publish the peer review history of their article (what does this mean?). If published, this will include your full peer review and any attached files.

Reviewer #1: **Yes: **Reyazul Rouf Mir

Reviewer #2: **Yes: **Sarfraz Ahmad

---

## [Author Response · Author response to Decision Letter 0]

3 Oct 2023

Responses to the Editor’s & Reviewer(s)’s Comments

#EDITOR’S COMMENTS

Comment#1: Please ensure that your manuscript meets PLOS ONE's style requirements, including those for file naming.

Reply: The manuscript meets PLOS ONE's style requirements

Comment#2: We suggest you thoroughly copyedit your manuscript for language usage, spelling, and grammar.

Reply: We have corrected the manuscript for language usage, spellings and grammar.

Upon resubmission, please provide the following:

Comment#2(i): The name of the colleague or the details of the professional service that edited your manuscript.

Reply: The manuscript is checked by all the authors of the manuscript. We did not hire any professional service because we do not have funds to pay for professional service.

Comment#2(ii): A copy of your manuscript showing your changes by either highlighting them or using track changes (uploaded as a *supporting information* file).

Reply: A copy of your manuscript showing your changes in track mode has been uploaded as “supporting information”.

Comment#2(iii): A clean copy of the edited manuscript (uploaded as the new *manuscript* file).

Reply: A clean copy of the edited manuscript has been uploaded as the new *manuscript* file.

Comment#3: Please note that funding information should not appear in any section or other areas of your manuscript. We will only publish funding information present in the Funding Statement section of the online submission form. Please remove any funding-related text from the manuscript.

Reply: Funding Information has been removed from the manuscript

Comment#4: We note that the grant information you provided in the ‘Funding Information’ and ‘Financial Disclosure’ sections do not match. 

Reply: Funding Information has been removed from the manuscript and information given in ‘Financial Disclosure’ section is corrected.

Comment#4(i): When you resubmit, please ensure that you provide the correct grant numbers for the awards you received for your study in the ‘Funding Information’ section.

Reply: We did not receive any specific funding for this study, therefore, there is no grant number to be indicated.

Comment#5: Thank you for stating the financial disclosure. Please state what role the funders took in the study. If the funders had no role, please state: "The funders had no role in study design, data collection and analysis, decision to publish, or preparation of the manuscript." 

Reply: Not applicable

Comment#6: We note that you have stated that you will provide repository information for your data at acceptance. Should your manuscript be accepted for publication, we will hold it until you provide the relevant accession numbers or DOIs necessary to access your data. If you wish to make changes to your Data Availability statement, please describe these changes in your cover letter and we will update your Data Availability statement to reflect the information you provide.

Reply: All data collected during the study is included in the manuscript. Even then, if you require any replication-wise raw data we will provide you after acceptance if need be.

Comment#7: PLOS requires an ORCID iD for the corresponding author in Editorial Manager on papers submitted after December 6th, 2016. Please ensure that you have an ORCID iD and that it is validated in Editorial Manager. To do this, go to ‘Update my Information’ (in the upper left-hand corner of the main menu), and click on the Fetch/Validate link next to the ORCID field. This will take you to the ORCID site and allow you to create a new iD or authenticate a pre-existing iD in Editorial Manager.

Reply: We will comply it while resubmission of the manuscript.

Comment#8: Please amend either the title on the online submission form (via Edit Submission) or the title in the manuscript so that they are identical.

Reply: Title is corrected in the online submission form.

Comment#9: Please include your full ethics statement in the ‘Methods’ section of your manuscript file. In your statement, please include the full name of the IRB or ethics committee who approved or waived your study, as well as whether or not you obtained informed written or verbal consent. If consent was waived for your study, please include this information in your statement as well. 

Reply: Not applicable to our study

Comment#10: Please review your reference list to ensure that it is complete and correct. If you have cited papers that have been retracted, please include the rationale for doing so in the manuscript text, or remove these references and replace them with relevant current references. Any changes to the reference list should be mentioned in the rebuttal letter that accompanies your revised manuscript. If you need to cite a retracted article, indicate the article’s retracted status in the References list and also include a citation and full reference for the retraction notice.

Reply: Roussel, 2005 was deleted as it was a mistake in year from page no.5. Actually the year was 2004.

Correct reference is:- 

Roussel V, Koenig J, Beckert M, Balfourier F. Molecular diversity in French bread wheat accessions related to temporal trends and breeding programmes. TheorAppl Genet. 2004; 108(5): 920–930. doi: 10.1007/s00122-003-1502-y

Following extra reference was also deleted:- 

Jasrotia P, Yadav, J, Kashyap PL, Bhardwaj AK, Kumar S, Singh GP (2020) Impact of Climate Change on Insect Pests of Rice–Wheat Cropping System: Recent Trends and Mitigation Strategies. In: Improving Cereal Productivity through Climate Smart Practices. Woodhead Publishing, Elsevier Inc. pp. 225-239

Comment#11: While revising your submission; please upload your figure files to the Preflight Analysis and Conversion Engine (PACE) digital diagnostic tool, https://pacev2.apexcovantage.com/. PACE helps ensure that figures meet PLOS requirements.

Reply: Ok we will comply to your suggestion. 

#REVIEWER 1

The manuscript entitled "Unravelling the novel genetic diversity and marker-trait associations of corn leaf aphid" is dealing with one of the most important insect of wheat. The study reports important marker trait associations genes for corn leaf aphid. The study is important and I recommend acceptance of this manuscript after the authors make following changes in the manuscript:

Comment#1: A few other aphid species also infest wheat crop in north-western plains of Punjab and Haryana. So, how you ensure that the standing wheat crop was only infested with R. maidis and no other aphid species were present?

Reply: The crop was sown in the screen house and a pure culture of R. maidis was maintained in the lab and released on the plants. The plants were checked thoroughly from time to time to ensure that species of aphid infesting the plants.

Comment#2: In the whole manuscript the ‘%’ sign should be replaced with ‘per cent’ except those in the brackets.

Reply: The sign ‘%’ is replaced with ‘per cent’.

Comment#3: On Page no. 4, Line no. 86, the sentence ‘DNA markers linked to insect resistance genes can be identified using Simple-Sequence Repeats (SSRs)’ needs to be revised because SSRs are itself DNA markers.

Reply: The sentence is changed as ‘The QTL’s linked to insect resistance…..’

Comment#4: On Page no. 4, Line no. 91, insert space between ‘breeding process’ and ‘(Wani et al. 2022)’.

Reply: Space provided.

Comment#5: What was the criteria for selection and why only these 48 wheat germplasms were taken into consideration?

Reply: Rigorous screening of around 1000 wheat genotypes for aphid resistance was carried out for 3 years. After screening, the promising genotypes with higher level of aphid resistance were selected. For comparison purpose, susceptible, moderately susceptible genotypes against aphids along with the popular genotypes among farmers were also included in the study.

Comment#6: At which time or stage or interval, the number of aphids infesting wheat plants was counted for phenotyping.

Reply: For phenotyping, the number of aphids infesting wheat plants was recorded three times during the experiment. The first observation was taken when the seedlings were 15 days old. The subsequent observations were taken at one month interval. All the aphid stages viz., nymphs, adults and alates (winged aphids) were taken into consideration.

Comment#7: On Page no. 10, Line no. 219, insert space between ‘-log10(p-value) ≤ 2’ and ‘To visualize’.

Reply: Space provided.

Comment#8: As per the manuscript, to visualize the results of association mapping, Manhattan plots were created. But why these are not included in the manuscript?

Reply: Corrections made.

Comment#9: On Page no. 19, Line no. 442, insert space between ‘aphid resistance.’ and ‘LD’.

Reply: Space provided.

#Reviewer 2

Comment#1: The title should give a good overview of the paper's focus. The present title is unclear and misleading about whether the work was done using corn or wheat. Authors must include the crop name in the title for clarity of the work.

Reply: As suggested, the crop name (wheat) is included in the title.

Comment#2: The use of SSR markers is appropriate for studying genetic diversity, but it would be beneficial to discuss why SSRs were chosen over other marker types (e.g., SNP markers) and their advantages and limitations in this context.

Reply: It is already discussed in the ‘Introduction’ section.

Comment#3: Ensure that the references cited in the manuscript are up-to-date and relevant to the topic and also correctly formatted.

Reply: As suggested, necessary changes have been incorporated in the references.References upto yaer 2022 and 2023 are covered in the manuscript.

---

## [Decision Letter · Decision Letter 1]

17 Oct 2023

Unravelling the novel genetic diversity and marker-trait associations of corn leaf aphid resistance in wheat using microsatellite markers

PONE-D-23-22687R1

Dear Dr. Jasrotia,

We’re pleased to inform you that your manuscript has been judged scientifically suitable for publication and will be formally accepted for publication once it meets all outstanding technical requirements.

Kind regards,

Mehdi Rahimi, Ph.D.

Academic Editor

PLOS ONE

Additional Editor Comments (optional):

Reviewers' comments:

Reviewer's Responses to Questions

**Comments to the Author**

1. If the authors have adequately addressed your comments raised in a previous round of review and you feel that this manuscript is now acceptable for publication, you may indicate that here to bypass the “Comments to the Author” section, enter your conflict of interest statement in the “Confidential to Editor” section, and submit your "Accept" recommendation.

Reviewer #2: All comments have been addressed

2. Is the manuscript technically sound, and do the data support the conclusions?

Reviewer #2: Yes

3. Has the statistical analysis been performed appropriately and rigorously? 

Reviewer #2: Yes

4. Have the authors made all data underlying the findings in their manuscript fully available?

Reviewer #2: Yes

5. Is the manuscript presented in an intelligible fashion and written in standard English?

Reviewer #2: Yes

6. Review Comments to the Author

Reviewer #2: (No Response)

7. PLOS authors have the option to publish the peer review history of their article (what does this mean?). If published, this will include your full peer review and any attached files.

Reviewer #2: **Yes: **Sarfraz Ahmad

---

## [Editor Report · Acceptance letter]

31 Oct 2023

PONE-D-23-22687R1 

Unravelling the novel genetic diversity and marker-trait associations of corn leaf aphid resistance in wheat using microsatellite markers 

Dear Dr. Jasrotia:

I'm pleased to inform you that your manuscript has been deemed suitable for publication in PLOS ONE. Congratulations! Your manuscript is now with our production department. 

Kind regards, 

on behalf of

Associate Prof. Mehdi Rahimi 

Academic Editor

PLOS ONE